# Clustering of Six Key Risk Behaviors for Chronic Disease among Adolescent Females

**DOI:** 10.3390/ijerph17197211

**Published:** 2020-10-02

**Authors:** Lauren A. Gardner, Katrina E. Champion, Belinda Parmenter, Lucinda Grummitt, Cath Chapman, Matthew Sunderland, Louise Thornton, Nyanda McBride, Nicola C. Newton

**Affiliations:** 1The Matilda Centre for Research in Mental Health and Substance Use, University of Sydney, Sydney 2006, Australia; katrina.champion@sydney.edu.au (K.E.C.); lucinda.grummitt@sydney.edu.au (L.G.); cath.chapman@sydney.edu.au (C.C.); matthew.sunderland@sydney.edu.au (M.S.); louise.thornton@sydney.edu.au (L.T.); nicola.newton@sydney.edu.au (N.C.N.); 2Department of Exercise Physiology, School of Medical Sciences, UNSW Sydney, Sydney 2052, Australia; b.parmenter@unsw.edu.au; 3National Drug Research Institute, Curtin University, Perth 6102, Australia; n.mcbride@curtin.edu.au; 4Weight Loss Center, Sacramento, CA 94203, USA; info@health4life.org.au

**Keywords:** chronic disease risk, females, multiple health behavior change, public health, physical activity, sleep, recreational screen time, diet, alcohol, smoking

## Abstract

Chronic diseases are the leading cause of disability and mortality globally. In Australia, females are at heightened risk. This research explored the prevalence, patterns, and correlates of six key risk behaviors (physical inactivity, poor diet, recreational screen time, inadequate sleep, alcohol use, and smoking) among adolescent females and whether knowledge of health guidelines was associated with adherence. Adolescent females completed an anonymous online questionnaire (*N* = 687; M_age_ = 13.82). Logistic regression assessed the association between knowledge and adherence. A Latent Class Analysis (LCA) and three-step procedure identified risk behavior clusters and their correlates. Despite positive health self-ratings (77% good/very good), most participants reported insufficient moderate-to-vigorous physical activity (MVPA; 89%), vegetable intake (89%), and excessive screen time (63%). Knowledge of guidelines was associated with adherence for MVPA, vegetable intake, sleep, and alcohol abstinence. Three classes emerged: “moderate risk” (76%), “relatively active, healthy eaters” (19%), and “excessive screen users” (5%). These risk-behavior clusters were associated with perceived value of academic achievement and physical wellbeing. Adolescent females commonly perceive they are in good health, despite engaging in unhealthy behaviors. Public health interventions should utilize effective behavior change strategies, adopt a multiple health behavior change approach (MHBC), and be tailored to specific risk profiles and values among females.

## 1. Introduction

Chronic diseases, such as cardiovascular disease, cancers, and mental disorders, are the leading cause of disability and mortality worldwide [1]. In Australia, almost 1 in 2 (49%) females have a chronic disease, with the risk of having one or more chronic conditions being higher among females than males [2]. Six key risk behaviors have been identified as increasing the risk of chronic disease, including smoking, alcohol use, physical inactivity, poor diet, sedentary behavior (e.g., recreational screen time), and poor sleep [3]. Importantly, these risk behaviors are modifiable, and therefore the associated disease and deaths are avoidable. Given that these six risk behaviors typically develop in adolescence, disease prevention efforts need to occur early. 

The six key risk behaviors differ among male and female adolescents. In Australia, for example, adolescent females are less likely to achieve the recommended amounts of physical activity and sleep, whereas adolescent males are more likely to engage in excessive screen time and use alcohol and tobacco [4,5,6]. Adolescent females are also more likely than males to experience symptoms of anxiety and depression, which are key short-term harms related to the six risk behaviors [7,8,9]. For example, screen time has been associated with depressive symptoms and lower self-esteem among females but higher self-esteem in males [10]. Similarly, evidence suggests obesity increases from childhood to late adolescence in females, whereas for males, this pattern is reversed [11]. As depression, anxiety, and obesity rates are continuing to rise among adolescent females globally, it is critical to better understand associated risk behaviors to aid prevention efforts and interrupt the long-term trajectory towards chronic disease [9,12,13].

A common strategy to address risk behaviors involves raising awareness of, and increasing compliance with, public health guidelines. In Australia, guidelines exist for many health behaviors, including physical activity, diet, sleep, screen time, and alcohol use. For example, it is recommended that children and young people (5–17 years) limit recreational screen time to 2 h or less per day [14]. Internationally, research investigating the link between knowledge and adherence to health guidelines has predominately focused on adults and has produced mixed results regarding physical activity [15] and diet [16]. Among an adolescent sample in Australia, knowledge of the physical activity guideline was not associated with adherence, and adolescents who knew the screen time guideline were less likely to adhere to it [17]. However, awareness of health guidelines is often low [16,17], and more research is needed to explore the links between knowledge and adherence to diet, sleep, and alcohol guidelines, particularly in adolescent female populations. Public health interventions are rarely gender-specific, despite evidence of key differences among males and females on the prevalence and impact of these risk factors.

It is also important to recognize that the six risk behaviors commonly co-occur, or ‘’cluster’’, with the likelihood of engaging in multiple risk behaviors increasing throughout adolescence [6,18]. Clustering of risk behaviors is problematic, as the risk of mortality increases beyond simply adding the effect of single behaviors [3,19]. This clustering is reflected in the recent adoption of 24-h movement guidelines in Canada [20] and Australia [14], which recognize the importance of addressing physical activity, sedentary behavior/screen time, and sleep in conjunction. Alcohol use and smoking are also known to commonly co-occur [21], and the clustering of physical activity, diet, and sedentary behavior in adolescence is well established [22]. Evidence suggests this clustering differs by sex, with adolescent females comprising unhealthy clusters characterized by low physical activity [22], and females more likely than males to engage in multiple risk behaviors [23]. Moreover, a recent review of school-based eHealth interventions targeting multiple health behaviors found adolescent females do not benefit as much as males in terms of physical activity [24], suggesting tailored interventions for females may be required. There is therefore a need to better understand the risk behavior profiles of adolescent females, along with their correlates, to inform future prevention work. To our knowledge, patterns and clustering of all six risk behaviors has not been explored in a sample of female adolescents.

This study aimed to conduct exploratory analyses to

Provide a snapshot of adolescent females’ perceptions of their health and their health behaviors, and investigate whether knowledge of health guidelines is associated with adherence;Explore patterns of clustering of the six risk behaviors and identify correlates of these behavioral profiles, including the perceived value of different life domains, to inform preventive intervention approaches.

## 2. Materials and Methods 

### 2.1. Participants and Procedure

Seven Australian independent secondary schools in New South Wales and the Australian Capital Territory of Australia were approached to participate in the study. Three schools agreed to participate. All students in Years 7–9 at participating schools were invited to participate. Passive parental consent and active student consent was gained from all participants included in the study (99% consent rate). Participants completed an anonymous online questionnaire (~20–30 min) during health education from August–September 2018. Respondents were given the opportunity to enter a draw to win a Fitbit (valued at 400AUD), with one prize allocated per school. The study was approved by the UNSW Sydney Human Research Ethics Committee (HC180224).

### 2.2. Measures

#### 2.2.1. Self-Rated Health

A single item assessed participants’ perceptions of their own health: “How would you rate your own health?”. Response options were “Excellent”, “Very Good”, “Good”, “Fair”, and “Poor”.

#### 2.2.2. Knowledge of Health Guidelines

Five multiple choice items assessed students’ knowledge of age-appropriate health guidelines. This included the Australian 24-h movement guidelines for children (5–12 years) and young people (13–17 years) [14], the Australian Dietary Guidelines [25], and the Australian guidelines to reduce health risks from drinking alcohol [26]. As there are no Australian health guidelines for smoking, participants’ knowledge against national guidelines could not be assessed. Knowledge items and correct responses are outlined in Appendix A, Table A1. Knowledge was represented by a dichotomous variable (0 = correct, 1 = incorrect).

#### 2.2.3. Moderate-Vigorous Physical Activity (MVPA)

Students reported the number of days they participated in at least 60 min/day of MVPA over a typical week [27].

#### 2.2.4. Sedentary Recreational Screen Time

A modified version of the Adolescent Sedentary Activity Questionnaire (ASAQ; [28]) assessed recreational screen time. The ASAQ has been found to be valid and reliable among adolescents [28]. Participants were instructed to “Think about a normal school week and record how long you believe you spend in recreational screen time before and after school each day” (in hours and minutes). The same question was asked in relation to ‘’a normal weekend’’. Responses exceeding 8hrs on a weekday and 16hrs on a weekend day were excluded (*n* = 92). Total weekly screen time was computed and averaged across the seven days.

#### 2.2.5. Fruit and Vegetable Intake

Two validated items commonly used in health research assessed fruit and vegetable consumption [29]. Fruit intake was assessed using “About how many serves of fruit do you usually eat each day?”, with response options ranging from “0 serves per day” to “6+ serves per day”. The same question was asked in relation to vegetables. Participants were provided with examples of what constitutes one serve.

#### 2.2.6. Sleep

Participants were asked “What time do you usually go to bed at night?” and “What time do you usually get up each morning?” (in hours and minutes, AM/PM). Self-reported estimates of bedtime, wake time, and sleep duration have been shown to be reliable and valid [30]. Total sleep duration was computed and responses with extreme values ( <3 h or >14 h) excluded from analyses (*n* = 15). Healthy sleep was defined as 9–11 h for 6–13 year-olds, and 8–10 h for 14–17 year-olds [14].

#### 2.2.7. Alcohol Use

Based on the Australian National Drug Strategy Household Survey [6], participants were asked “Have you had a standard alcoholic drink (e.g., 100 mL of wine, mid strength beer) in the past 6 months?” (Yes/No).

#### 2.2.8. Smoking

Based on the Standard High School Youth Risk Behavior Survey [31], participants were asked “Have you ever tried cigarette smoking, even one or two puffs?” (Yes/No).

#### 2.2.9. Adherence to Health Guidelines

Adherence to physical activity, screen time, diet, sleep, and alcohol-use guidelines was assessed using the measures described above and represented by a dichotomous variable (0 = adherence to the guideline, 1 = failing to meet guideline).

#### 2.2.10. Perceived Value of Life Domains

Participants ranked five life domains in order of importance (1 = Most Important, 5 = least important): physical wellbeing, mental wellbeing, social relations, familial connections, and academic achievement.

### 2.3. Statiscial Analysis

Descriptive statistics and logistic regression analyses were conducted in SPSS Statistics Software (version 24) (IBM Corp., New York, USA). Logistic regression assessed whether knowledge of health guidelines was associated with adherence, or in the case of alcohol where a specific guideline is not available, having had a standard alcoholic drink in the prior 6 months. A Latent Class Analysis (LCA) was performed using Mplus version 8 [32] to identify clusters of health behaviors. Continuous indicators included the typical number of days per week where 60 min of MVPA was achieved, average daily recreational screen time, and daily serves of fruit and vegetables. Categorical indicators included the consumption a standard alcoholic drink in the prior 6 months, having ever tried smoking, and healthy sleep (i.e., falling within the healthy sleep range). The optimal number of latent classes was informed by running sequential LCA models, each time increasing the number of classes by one. Model fit criteria, including the Akaike information criterion (AIC) and sample-size-adjusted Bayes information criterion (aBIC), were inspected with lower values, indicating improved model fit. Lo–Mendell–Rubin Adjusted Likelihood Ratio Tests (LMRT) and Bootstrap Likelihood Ratio Tests (BLRT) were used to examine comparative model fit. The LMRT and BLRT p-values indicate whether the model with more classes (*p* > 0.05) or less classes (*p* < 0.05) is optimal. Relative entropy and size and interpretability of latent classes were also considered. Finally, the three-step procedure explored class differences in the ranked value of life domains while adjusting for classification errors [33].

## 3. Results

### 3.1. Self-Rated Health; Risk Behaviors; and Knowledge of, and Adherence to, Health Guidelines

A total of 687 females (Mage = 13.82 years, SD = 0.88) completed the survey (see Table 1). Most participants perceived their health to be ‘’very good’’ (42%) or ‘’good’’ (35%). Despite this, adherence to guidelines was poor for some behaviors, with 89% of the sample not engaging in sufficient MVPA or eating enough vegetables, 63% engaging in excessive recreational screen time, and 40% getting inadequate sleep. One-fifth of the sample did not eat enough fruit (20%), 14% had consumed a standard alcoholic drink within the prior 6 months, and 2% had tried tobacco. Knowledge of health guidelines was high for alcohol (94%) and sleep (71%), but limited for screen time (52%), fruit (37%) and vegetable (35%) intake, and physical activity (26%).

### 3.2. Logistic Regression

Logistic regression analyses indicated that knowledge of the health guideline was associated with greater adherence for MVPA (OR = 3.89, 95% CI [2.39–6.35], *p* < 0.001), vegetable intake, 95% CI [2.31–6.33], *p* < 0.001), and sleep (OR = 1.42, 95% CI [1.11–1.83], *p* = 0.006). Additionally, knowledge of the alcohol guideline was positively associated with not having consumed a standard alcoholic drink in the prior 6 months (OR = 3.66, 95% CI [1.61–8.33], *p* = 0.002). Fruit intake knowledge was inversely associated with adherence (OR = 0.53, 95% CI [0.358–775], *p* = 0.001), and knowledge of the recreational screen time guideline was not associated with adherence (OR = 1.04, 95% CI [0.76–1.42], *p* = 0.797).

### 3.3. Latent Class Analysis 

The results of the LCA are presented in Table 2 (*N* = 675). Six cases had missing data on all variables and were excluded. Although the LMRT value within the two-class solution was not significant, the three-class solution was deemed superior to the two-class solution due to significant LMRT, BLRT, and lower AIC and aBIC values. The four-class solution revealed slightly higher entropy, lower AIC and aBIC, and a significant BLRT value; however, the LMRT indicated that it was not statistically different from the three-class solution. Additionally, one of the classes in the four-class model was comprised of only 13 participants (1.93%) and was considered too small. The three-class solution was therefore selected as the best fitting model (see Figure 1). Average probabilities for the most likely latent class membership ranged between 0.84 and 0.93. 

Class 1 (*n* = 513; 76%) was labelled “moderate risk”, as, similar to Class 3, it was characterized by engaging in the least amount of recreational screen time (2.75 h/day; *SEM* = 0.10) despite still exceeding the health guideline, and a moderate probability of healthy sleep (64%; *SEM* = 0.03). This class had the lowest vegetable intake (2.44 serves/day; *SEM* = 0.09); however, the guideline for fruit intake was achieved (2.17 serves/day; *SEM* = 0.07). Compared to Classes 2 and 3, Class 1 participated in a moderate amount of MVPA (4.18 days/week; *SEM* = 0.10), although still did not meet recommendations. As with each of the three classes, this class was characterized by a low probability of drinking alcohol and smoking.

Class 2, the smallest class (*n* = 36; 5.33%), was labelled “excessive screen users” due to being characterized by high amounts of recreational screen time (8.39hrs/day; *SEM* = 0.92), the least MVPA (3.47 days/week; *SEM* = 0.31), and the least likely to achieve healthy sleep (37%; *SEM* = 0.09). Although this class achieved the recommended fruit intake (2.92 serves/day; *SEM* = 0.41), they had low vegetable intake (2.48 serves/day; *SEM* = 0.29).

Class 3 (*n* = 126; 18.67%) was labelled “relatively active, healthy eaters” due to participating in the most MVPA (4.95 days/week; *SEM* = 0.18) and eating the most fruit (4.49 serves/day; *SEM* = 0.28) and vegetables (3.94 serves/day; *SEM* = 0.15). However, the term “relatively” is used as the class still fell below the recommended level of MVPA and vegetable intake. Similar to Class 1, this class engaged in the least amount of recreational screen time (2.76 h/day; *SEM* = 0.19), although still exceeded the guideline of 2 h, and had a moderate probability of getting adequate sleep (54%; *SEM* = 0.06).

### 3.4. Asssociation of Latent Classes with Ranked Value of Life Domains

There was no difference between classes on ranked value of mental wellbeing, social relations, or familial connections. However, a difference was found for ranked value of academic achievement. Specifically, Class 1 (“moderate risk”) and Class 2 (“excessive screen users”) were significantly more likely to rate academic achievement as more important than Class 3 (“relatively active, healthy eaters”) (OR = 1.20, 95% CI [1.00–1.43], *p* = 0.027, OR = 1.73, 95% CI [1.21–2.49], *p* < 0.001, respectively), whereas Class 1 (“moderate risk”) was significantly less likely to rate academic achievement as more important than Class 2 (“excessive screen users”) (OR = 0.69, 95% CI [0.49–97], *p* = 0.009). The classes also differed on ranked value of physical wellbeing, with Class 1 (“moderate risk”) and Class 3 (“relatively active, healthy eaters”) significantly more likely to rate physical wellbeing as more important than Class 2 (“excessive screen users”) (OR = 1.29, 95% CI [0.99–1.68], *p* = 0.033, OR = 1.48, 95% CI [1.08–2.03], *p* = 0.046, respectively).

## 4. Discussion

This study explored adolescent females’ perceptions of their health, and their knowledge of and adherence to Australian health guidelines, along with clusters and correlates of six key risk behaviors for chronic disease: physical inactivity, poor diet, poor sleep, excessive recreational screen time, alcohol use, and smoking. Despite having positive perceptions of their health, the majority of participants reported insufficient MVPA (89%), and vegetable intake (89%), and engaged in excessive recreational screen time (63%). Forty percent had inadequate sleep, 20% did not eat enough fruit, 14% consumed alcohol within the prior 6 months, and 2% had tried tobacco. Knowledge of guidelines was associated with adherence to the guidelines for MVPA, vegetable intake, and sleep, and was associated with abstaining from alcohol. Finally, distinct clusters of behaviors were evident, and these clusters were associated with how highly individuals valued their academic achievement and physical wellbeing.

The current samples’ positive health self-ratings align with other Australian data that found 67% of females aged 15–24 rated their health as excellent or very good, with younger females typically having more positive perceptions of their health [34]. The incongruence between self-rated health and prevalence of risk behaviors may reflect adolescent females’ conceptualizations of health. Whilst adults tend to base their perceptions of health on the presence or absence of illness, adolescents are typically free of disease and may consider aspects beyond physical health determinants [35]. For example, a recent qualitative analysis found adolescents’ descriptions of health centered on physical appearance, personal commitment and goals, possessions and space, use of free time, and social belonging [36]. Nevertheless, the finding that adolescent females reported poor MVPA, low vegetable intake, high recreational screen time, and poor sleep is important and aligns with other Australian reports that found, among adolescent females, 93% do not meet the MVPA guideline, 89% do not eat enough vegetables, 74% exceed the screen time guideline, and 30% get insufficient sleep [4,5,37]. Identified rates of fruit intake, alcohol use, and smoking are also consistent with Australian data [6,17]. The fact that 14% of the sample have consumed alcohol is worth noting within the context of research across several countries that has found the gap between male and female alcohol use is lessening among recently born cohorts [38]. This may be due to males and females having unique neurobiological, psychiatric, and social vulnerabilities to alcohol use, highlighting the need to examine and target these groups separately [39].

With the exception of sleep and alcohol, knowledge of health guidelines was poor. This is consistent with the limited research among Australian adolescents that found only 24% knew the physical activity guideline and 9.5% knew the screen time guideline [17]. Hardy, Mihrshahi, Bellew, Bauman, and Ding [17] did not find a link between knowledge and adherence to the physical activity guideline, and knowledge of the screen time guideline was associated with a reduced adherence. However, this study did not report data separately for males and females, making direct comparison with the current sample difficult. Despite the current research finding a positive association between knowledge and adherence to guidelines for MVPA, vegetable intake, and sleep, there was an inverse association with fruit intake. Overall, these equivocal findings suggest a tenuous link between knowledge and adherence to health guidelines. While it is evident that there is a need to improve knowledge of health guidelines, awareness of recommendations alone may not be enough to elicit behavior change.

The identified clustering of physical activity, diet, and screen time is consistent with previous findings among children and adolescents [22]. In particular, among six identified obesogenic behavior clusters in adolescent females, Boone–Heinonen et al. [40] similarly found a cluster characterized by moderate levels of physical activity, fruit/vegetable consumption, and screen use; another characterized by high levels of screen use; and another characterized by the greatest intake of fruit and vegetables along with relatively high engagement in sport and exercise. The pattern of high screen time with low MVPA and poor sleep in Class 2 (“excessive screen users”) supports the new 24-h movement guidelines [14,20], which highlight that within a given 24-h period, time spent on one behavior displaces time spent on one of the other behaviors [14].

The links between the risk clusters and perceived value of different life domains is novel. Interestingly, Class 2 (“excessive screen users”) valued academic achievement more highly than the other two classes did, while research suggests less screen time and better sleep are linked to higher academic performance [41]. However, more research is needed to understand whether beliefs about the importance of academic achievement predict academic performance among female adolescents. Class 2 (“excessive screen users”) also valued physical wellbeing less than the other two classes did, suggesting a lack of understanding of the importance of physical wellbeing for academic achievement, and in turn, the impact of high screen time and poor sleep on both physical wellbeing and academic achievement [42]. However, despite valuing physical wellbeing highly, Class 1 (“moderate risk”) and Class 3 (“relatively active, healthy eaters”) were not meeting guidelines for many health behaviors. Nevertheless, understanding what the different classes value most provides additional information that can be used to inform and tailor preventive interventions to increase engagement and effectiveness. For example, Class 2 may benefit most from content linking health to academic achievement, whereas Classes 1 and 3 might benefit more from content and goals linked to improving their physical wellbeing.

### 4.1. Practical Implications for Public Health

The present findings highlight the high prevalence of unhealthy behaviors among adolescent females, along with the lack of knowledge of health guidelines, suggesting a need for improved health promotion and preventive interventions among this group. In addition to improving knowledge of guidelines, preventive public health interventions should incorporate other effective behavior-change strategies, such as practicing skills for good health (e.g., self-monitoring and goal setting), and providing normative education [43,44,45,46]. Further, the clustering of behaviors suggests that interventions that adopt a multiple health behavior change (MHBC) approach [47], whereby health behaviors are targeted together, rather than in isolation, may be beneficial among this population. However, given that clustering of health behaviors differs by sex [22], that adolescent females and males may respond differently to MHBC interventions [24], and that the clusters in the current study varied in perceived importance of life domains, considerations should be given to sex, risk profile, and value alignment when developing interventions. One approach involves utilizing web- and mobile-based technologies, which can be particularly useful for tailoring interventions [24]. Based on the present risk profiles and correlates, another idea would be to provide information within study skill sessions for females to highlight the important link between physical wellbeing, particularly in regard to healthy sleep and limited screen time, and academic achievement.

### 4.2. Limitations and Future Directions

The current study is the first among Australian adolescent females to explore whether knowledge of health guidelines is associated with adherence to the guidelines for MVPA, fruit and vegetable intake, recreational screen time, sleep, and alcohol use, along with clustering and correlates of the six risk behaviors. Given the data are cross-sectional in nature, causation and directionality of relationships cannot be inferred. Another limitation is the reliance on self-report measures. Future research may benefit from using objective measures, such as accelerometers, which can provide accurate measurements of physical activity, sedentary behavior, and sleep. It may also be useful to consider other indicators of each risk behavior, for example, sugar-sweetened beverage consumption, and sleep quality and patterns. Future research exploring the link between knowledge and adherence to guidelines should control for other factors (e.g., physical health conditions) that could influence the relationship and potentially moderating variables, such as parental decision making and style. Moreover, research could consider how the risk clusters relate to other factors such as obesity and mental health. Finally, the present research only included participants from independent schools in metropolitan regions of a relatively high socioeconomic status (SES). Given low SES has been linked to poorer health outcomes among adolescents [48], future research should include participants from a range of socioeconomic positions, school types, and geographic locations.

## 5. Conclusions

Despite having positive perceptions of their own health, many adolescent females engage in multiple unhealthy behaviors and lack knowledge of health guidelines, putting them at risk of both short-term health problems and chronic disease in adulthood. This is important given the higher rates of anxiety and depression among adolescent females compared to males, and the greater risk of having a chronic condition in adulthood. Although there is some evidence suggesting a link between knowledge of health guidelines and health behaviors, public health interventions need to go beyond merely raising awareness of guidelines and utilize effective behavior change strategies (e.g., self-monitoring, goal-setting, and normative education). Distinct patterns of risk behaviors exist among adolescent females and relate to differences in perceived importance of life domains, highlighting the potential value of MHBC preventive interventions that are tailored to females, their risk profile, and their value alignment.

## Figures and Tables

**Figure 1 ijerph-17-07211-f001:**
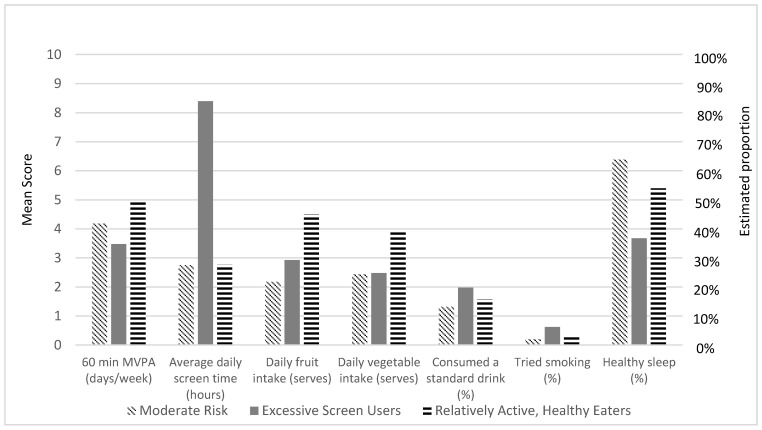
Mean scores and estimated proportion for health behaviors in each latent class.

**Table 1 ijerph-17-07211-t001:** Descriptive statistics and prevalence estimates.

	Behaviour		Knowledge of Guidelines		Adherence to Guidelines	
	Mean	SD	*N*	%	*N*	%
MVPA for at least 60 min	4.30 (days/week)	1.77	175	25.70	76	11.3
Recreational Screen time	3.06 (h/day)	1.93	352	51.80	246	36.6
Fruit intake	2.69 (serves/day)	1.40	247	36.50	541	80.50
Vegetable intake	2.75 (serves/day)	1.32	237	35.00	74	11.00
Sleep	8.63 (h/night)	1.08	482	70.90	382	60.40
Alcohol	14.1% (*n* = 59) consumed a standard drink	-	633	93.60	360	85.90
Smoking	2.4% (*n* = 16) tried smoking	-	-	-	-	-

Note: MVPA guideline = 60 min on 7 days per week; screen time guideline = ≤2 h per day; fruit intake guideline = ≥2 serves per day; vegetable intake guideline = ≥5 serves per day; alcohol guideline = the safest option is not to drink; smoking guideline = N/A; sleep guideline = 9–11 h between 6–13 years of age and 8–10 h between 14–17 years of age.

**Table 2 ijerph-17-07211-t002:** Model fit indices for the latent class analysis.

Classes	AIC	aBIC	Entropy	H0 Loglikelihood Value	LMRT (*p*-Value)	BLRT *p*-Value
1	11,485.354	11,500.090	-	−5731.677	-	-
2	11,318.814	11,344.267	0.926	−5640.407	179.103 (*p* = *0*.384)	<0.001
3 *	11,192.008	11,228.177	0.800	−5569.004	140.118 (*p* < *0*.001)	<0.001
4	11,105.894	11,152.780	0.809	−5517.947	100.192 (*p* = *0*.212)	<0.001

Note: AIC = Akaike Information Criterion, aBIC = sample size adjusted Bayesian Information Criterion, LMRT = Lo–Mendell–Rubin Adjusted Likelihood Ratio Tests, BLRT = Bootstrap Likelihood Ratio Tests, * the three-class solution was retained.

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
