# Peer review of "Clustering of Six Key Risk Behaviors for Chronic Disease among Adolescent Females"

_ijerph, 2020, doi:10.3390/ijerph17197211_

Round 1

Reviewer 1 Report

This manuscript by Gardner et al. looked at the clustering of six key behaviours for chronic disease among adolescent females in Australia. The manuscript addresses an important health issue in an important high-risk group i.e. adolescents. The paper is well written, and the study was conducted well with a high response rate. I only have a few minor comments for the authors to consider.

  1. Where exactly the study was conducted in Australia?
  2. Four out of seven schools that were approached did not participate. Are there any school-level information available to compare the schools that did participate with those that did not for a possibility of selection bias?
  3. What were the variables adjusted for in the logistic regression analysis?
  4. Why drug abuse was not considered?
  5. Please check Ref 3 - multiple authors are listed.

Author Response

Response to Reviewer 1

Reviewer comment: This manuscript by Gardner et al. looked at the clustering of six key behaviours for chronic disease among adolescent females in Australia. The manuscript addresses an important health issue in an important high-risk group i.e. adolescents. The paper is well written, and the study was conducted well with a high response rate. I only have a few minor comments for the authors to consider.

  1. Where exactly the study was conducted in Australia?

Author comment: We thank the reviewer for their time reviewing the manuscript. The study was conducted in New South Wales and the Australian Capital Territory of Australia. This information has been added on page 3, lines 95-96.

Reviewer comment:

  1. Four out of seven schools that were approached did not participate. Are there any school-level information available to compare the schools that did participate with those that did not for a possibility of selection bias?

Author comment: Typically, participation rates in our large school-based trials are approximately 15%, meaning it is common for us to invite schools and have them choose not to participate. We do not collect information about the schools that choose not to participate. In the present study, all invited schools were independent schools within metropolitan regions, limiting the potential for selection bias. However, we recommend future research include participants from a range of socioeconomic positions, school types, and geographic locations (page 9, lines 341-344).

Reviewer comment:

  1. What were the variables adjusted for in the logistic regression analysis?

Author comment: No covariates were included in the logistic regression analyses. Given the sample was comprised only of females within a limited age range, and all participants were from independent schools in metropolitan regions, there was limited variance to control for. However, we acknowledge that future research could focus on a broader demographic on page 9, lines 341-344.

Reviewer comment:

  1. Why drug abuse was not considered?

Author comment: We chose to focus specifically on the six health behaviours due to evidence establishing them as key risk factors for chronic disease and mortality, with alcohol and tobacco considered traditional risk factors due to their well-known links to chronic disease (Ding et al., 2015). Links to chronic disease on a population level are based on prevalence, and alcohol and tobacco use are the most prevalent forms of substance use among Australians (Australian Institute of Health and Welfare, 2017; Guerin & White, 2018). Rates of other substance use (e.g., ecstasy, amphetamines, opiates etc.) are particularly low among younger adolescents (i.e., the age of the current sample) (Australian Institute of Health and Welfare, 2017; Guerin & White, 2018). As such, other substance use or abuse was not considered in this study.

Ding, D., Rogers, K., van der Ploeg, H., Stamatakis, E., & Bauman, A. E. (2015). Traditional and Emerging Lifestyle Risk Behaviors and All-Cause Mortality in Middle-Aged and Older Adults: Evidence from a Large Population-Based Australian Cohort. PLOS Medicine, 12(12), e1001917.

Australian Institute of Health and Welfare. (2017). National Drug Strategy Household Survey 2016: detailed findings. Canberra: AIHW

Guerin, N., & White, V. (2018). ASSAD 2017 Statistics & Trends: Australian Secondary Students’ Use of Tobacco, Alcohol, Over-the-counter Drugs, and Illicit Substances. Cancer Council Victoria

Reviewer comment:

  1. Please check Ref 3 - multiple authors are listed.

Author comment: Reference 3 has been checked and we confirm that it has been referenced correctly.

Reviewer 2 Report

This is a well put together article that presents an interesting differentiation of teenage risk groups.

I have only one significant piece to recommend

  • The abstract states that this research explored “whether knowledge of health guidelines predicts adherence.” Predicts is also used through the article and has a connotation of future events. As this is a cross sectional study, it is more that it explored whether knowledge of health guidelines correlates with adherence. Use "correlates with" instead of "predicts".

Other than this there are just a few things to define:

  • mage on line 175 not defined.
  • M in Table 1 is not defined
  • What does the * in Table 2 mean? I know you are trying to note which class size you have in the study but it is not specifically noted in the table legend.

Author Response

Response to Reviewer 2

Review comment: This is a well put together article that presents an interesting differentiation of teenage risk groups.

I have only one significant piece to recommend

The abstract states that this research explored “whether knowledge of health guidelines predicts adherence.” Predicts is also used through the article and has a connotation of future events. As this is a cross sectional study, it is more that it explored whether knowledge of health guidelines correlates with adherence. Use "correlates with" instead of "predicts".

Author comment: We thank the reviewer for their time reviewing this manuscript and raising this point. We have updated all instances of “predicts” to “associated with”.

Review comment: Other than this there are just a few things to define:

mage on line 175 not defined.

Author comment: Abbreviation has been spelt out as “Mean age”.

Reviewer comment: M in Table 1 is not defined

Author comment: “M” has been spelt out as “Mean”

Reviewer comment:What does the * in Table 2 mean? I know you are trying to note which class size you have in the study but it is not specifically noted in the table legend.

Author comment: Information has been added to the note in Table 2 to explain that the asterisk signifies that the three-class solution was retained.

Reviewer 3 Report

The article explores the prevalence, patterns, and correlates of six key risk behaviors (physical inactivity, poor diet, recreational screen time, inadequate sleep, alcohol use, smoking) among adolescent females and whether knowledge of health guidelines predicts adherence. Implications in the article are that public health interventions should utilize effective behavior change strategies, adopt a multiple health behavior change approach, and be tailored to specific risk profiles and values among females.

Strength of the article is that it presents many dependencies between health-related facts. I didn't notice any remarkable weaknesses.

Things to improve:

In Table 2 you mark that 3 classes is the optimal number of classes. However, AIC and aBIC don't have minimum at 3. This suggests that you have selected 3 classes based on Entropy which has minimum at three. If this is the case, you should write the entropy value 0.800 with bold characters to make clear why you chose 3 classes as the optimum. If this is not the case, make clear to what your selection is based on.

Reviewer 4 Report

I recommend the publication of this very well written paper

Author Response

Response to Reviewer 4

Review comment: I recommend the publication of this very well written paper.

Author comment: We thank the reviewer for taking the time to review our manuscript and recommending it for publication.